# The Effect of Subliminal Electrical Noise Stimulation on Plantar Vibration Sensitivity in Persons with Diabetes Mellitus

**DOI:** 10.3390/biomedicines10081880

**Published:** 2022-08-04

**Authors:** Tina J. Drechsel, Claudio Zippenfennig, Daniel Schmidt, Thomas L. Milani

**Affiliations:** 1Department of Human Locomotion, Faculty of Behavioral and Social Sciences, Institute of Human Movement Science and Health, Chemnitz University of Technology, 09107 Chemnitz, Germany; 2Motor Control, Cognition and Neurophysiology, Faculty of Behavioral and Social Sciences, Institute of Human Movement Science and Health, Chemnitz University of Technology, 09107 Chemnitz, Germany

**Keywords:** vibration perception threshold, mechanoreceptors, subliminal electrical noise stimulation, diabetes mellitus

## Abstract

Subliminal electrical noise (SEN) enhances sensitivity in healthy individuals of various ages. Diabetes and its neurodegenerative profile, such as marked decreases in foot sensitivity, highlights the potential benefits of SEN in such populations. Accordingly, this study aimed to investigate the effect of SEN on vibration sensitivity in diabetes. Vibration perception thresholds (VPT) and corresponding VPT variations (coefficient of variation, CoV) of two experimental groups with diabetes mellitus were determined using a customized vibration exciter (30 and 200 Hz). Plantar measurements were taken at the metatarsal area with and without SEN stimulation. Wilcoxon signed-rank and t tests were used to test for differences in VPT and CoV within frequencies, between the conditions with and without SEN. We found no statistically significant effects of SEN on VPT and CoV (*p* > 0.05). CoV showed descriptively lower mean variations of 4 and 7% for VPT in experiment 1. SEN did not demonstrate improvements in VPT in diabetic individuals. Interestingly, taking into account the most severely affected (neuropathy severity) individuals, SEN seems to positively influence vibratory perception. However, the descriptively reduced variations in experiment 1 indicate that participants felt more consistently. It is possible that the effect of SEN on thick, myelinated Aβ-fibers is only marginally present.

## 1. Introduction

Electrical noise is a physiological phenomenon that occurs in the central and peripheral nervous system of humans and other living organisms. Especially in thin and short nerve fibers (e.g., C-fibers, cerebellar parallel fibers, non-myelinated pyramidal cell axon collaterals), so-called channel noise leads to membrane potential variation due to the small axonal diameter. This enables action potentials to be transmitted randomly and spontaneously without external synaptic inputs [1].

Based on this physiological phenomenon, the effect of externally applied noise signals, called stochastic resonance, on different sensory systems of the human body has been investigated for several years. Subthreshold noise can improve the capability to detect weak stimuli or enhance the information content of a signal [2,3,4,5]. The noise signal can be applied mechanically or electrically. Two possible mechanisms of action underlie stochastic resonance. First, noise can affect the permeability of the cell membrane of the receptors and increase the sensitivity of the receptors by changing the membrane potential toward the depolarization threshold [6,7]. Second, noise can act directly on the nerve fiber [4,6]. If a subthreshold tactile stimulus occurs (e.g., a vibration of defined frequency), the stimulus and the noise signal overlap. Consequently, the stimulus exceeds the perception threshold and becomes perceptible to the participant [3,4,8].

Diabetes mellitus is a disease that may lead to degeneration processes in nerve fibers [9,10,11], as well as structural and quantitative changes in mechanoreceptors [9,10,11,12]. Consequently, somatosensation can deteriorate [13], which is an enormous risk factor for the development of diabetic foot ulcerations [14]. In the clinical setting, measuring the plantar vibration perception threshold (VPT) is considered an important indicator for detecting patients at risk for developing diabetic foot ulcers [15]. Conversely, improving VPT could possibly delay the development of diabetic foot ulceration. While it has already been demonstrated in healthy participants of different ages that sensory perception can be improved by means of SEN stimulation [4,6,16,17], to the best of the authors’ knowledge proof of efficacy in patients with diabetes mellitus is still lacking. Therefore, the present study aimed to investigate possible effects of SEN on plantar VPT in a cohort of participants with diabetes mellitus. By using more direct stimulation in contrast to a previous study of our working group [18], we hypothesize a positive effect of SEN stimulation on VPT in patients with diabetes mellitus.

## 2. Materials and Methods

### 2.1. Participants

This cross-sectional study investigated the effect of SEN stimulation on plantar vibration sensitivity in a total of 70 individuals with diabetes mellitus. Due to missing values, five participants were excluded from experiment 1 and six from experiment 2. The anthropometric data of the remaining 59 participants are presented in Table 1. 

All measurement procedures were performed based on recommendations of the Declaration of Helsinki and approved by the local ethics committee of the Chemnitz University of Technology (protocol V-379-17-TM-Inside-09042020). All subjects were informed in detail about the content and aims of the study and gave their written informed consent before participation. 

The subjects were recruited through two diabetic primary care practices in Chemnitz, based on the following inclusion criteria: age 18 years and over; absence of pregnancy; absence of malignant disease within the last five years; absence of diseases with a prognosis of less than five years, and severe diseases that do not allow participation according to the assessment of the study physician; absence of serious neurological impairments, e.g., multiple sclerosis, Parkinson’s disease; absence of severe cardiac arrhythmia and no use of cardiac pacemaker; absence of injuries to the lower extremities in the past six months; absence of thrombosis or hemophilia, clinical leg edema, and symptomatic peripheral arterial occlusive disease ≥ stage two (e.g., patients experience pain when walking); ability to communicate (e.g., no dementia, no pronounced hearing loss.); absence of neuropathies originating from diseases other than diabetes; no skin irritations, open skin areas, or burns, etc. in the area of the foot and lower leg, and no skin diseases caused by viruses, fungi, or bacteria; absence of acute dizziness, fever, infectious diseases, and medication affecting the central nervous system (e.g., opiate therapy); and no alcohol and drug consumption within the last 24 h. All recruited participants were patients of the two diabetologists, so the exclusion criteria were checked by the diabetologists based on experience and patient chart.

### 2.2. Measurement Procedure and Devices

Gaussian white noise (frequency range between 5 Hz and 2000 Hz [19], Butterworth filter first order) was applied through plate electrodes using an isolated bipolar constant current stimulator (Digitimer DS5, Welwyn Garden City, Hertfordshire, United Kingdom). Following Iliopoulos et al. (2014), the electrical noise was generated as an analog voltage signal (National Instruments (NI) 6211 data acquisition (DAQ) card), using a self-written LabVIEW program. The analog outputs were channeled into a Digitimer DS5, which converts the voltage signal into current [4]. The electrical noise was increased in 0.5 milliampere (mA) increments from zero to the point at which individuals felt a slight tingling sensation. After that, the noise was reduced again in 0.1 mA steps until the participants could no longer feel any tingling [6,19]. This signal level was defined as the individual current perceptual threshold [6,19]. The mean of three current perceptual thresholds was used to determine the stimulation intensity of 90% of the individual current perceptual threshold [20,21,22]. Two electrodes, one on the plantar surface of the metatarsal heads and one on the dorsal surface of the forefoot, (see Figure 1), were inserted into saltwater-soaked sponges and fixed in a stable position using elastic Velcro straps. The investigators paid special attention to good skin contact of the electrodes and a comfortable position of the electrodes at the measurement site.

In both experiments, VPT were assessed using a customized vibration exciter (Brüel & Kjaer Vibro GmbH; type 4180, Darmstadt, Germany). The plastic probe of the vibration exciter (diameter 7.8 mm) was positioned perpendicularly to the plantar measurement locations and supported by a swivel arm. The pressure of the probe against the measurement points was monitored, keeping it within a range of 0.7 N to 1.2 N [23]. Participants wore noise-cancelling headphones (QuietComfort 25, Bose GmbH, Friedrichsdorf, Germany) to eliminate environmental noises. Variations of room temperature and the temperature of the plantar surface were kept to a minimum (mean ± SD difference of foot temperature pre vs. post: −1.9 ± 2.1 °C, mean difference of room temperature pre vs. post: −0.9 ± 0.7 °C).

For experiment 1, VPT were assessed under the first metatarsal head (MTH1) of the left foot. After a ten-minute acclimatization period, each participant went through two randomized measurement blocks: a 200 Hz vibratory stimulation (MTH1_200 Hz) to target the Pacinian corpuscles [24,25] and a 30 Hz vibratory stimulation (MTH1_30 Hz) to target the Meissner corpuscles [26]. An additional practice trial was done at the beginning of each session. Each block consisted of six VPT trials with three measurement runs each with and without SEN. The order of the trials was randomized and blinded for the subjects. A self-written LabVIEW program ran a customized VPT protocol inspired by Mildren et al. [27] that applies several sinusoidal vibration bursts (two seconds duration followed by a two to seven seconds pause) per trial [23], with the participants pressing a button as soon as they felt the probe vibrate. The mean of the last recognized and the last unperceived vibration stimulus was determined as VPT [13]. SEN stimulation was performed parallel to the measurement of the VPT. The probe of the vibration exciter was applied directly to the skin at the measurement site through a hole in the plantar plate electrode and sponge (Figure 1a). For experiment 2, VPT were assessed under the third metatarsal head (MTH3) of the left foot using the same customized vibration exciter as in experiment 1. The plastic probe was applied externally to the plantar electrode and VPT was thus recorded indirectly through the plate electrode surrounded by the sponge (Figure 1b). As in experiment 1, each participant went through an additional practice trial at the beginning of the session and two randomized measurement blocks (MTH3_200 Hz & MTH3_30 Hz) consisting of six trials with three measurement runs each with and without SEN.

### 2.3. Statistical Analysis

VPT are recorded on a ratio scale, which may result in heteroscedastic and non-normal distribution [28]. To correct for this distribution, the VPT data can be transformed using the natural logarithm [28]. The extent to which logarithmization has improved the shape of the distribution can be descriptively tested using Bland–Altmann plots and by calculating Spearman’s correlation. Only 50% of the available data benefited from logarithmization, so the raw data were used for all further analyses. The mean values of three VPT trials per measurement block were used for statistical analysis. Wilcoxon signed-rank (not normally distributed data) and dependent t tests (normally distributed data) were used to test for differences in VPT within frequencies, and between the conditions with and without SEN (alpha = 0.05). We also calculated the coefficient of variation (CoV) of the VPT to represent individual variations and tested this parameter using the Wilcoxon signed-rank test for differences between the two measurement conditions. All statistical analyses were performed in R (version 3.6.3) [29].

Statistically significant differences between experiment 1 and 2 were found for HbA1c (*p* = 0.01, see Table 1). Since the study design did not include comparisons between both experiments, this difference is considered not relevant. Furthermore, a comparison of the data from both experiments was ruled out for two central reasons: first, the measurements within the two experiments were performed on different cohorts with different baseline values. Second, the measurement of VPT was performed at different measurement points (MTH1 vs. MTH3) with different receptor densities and distributions [30], and differently regarding the direct (experiment 1) and indirect (experiment 2) application of the vibration exciter. This is also reflected in the data of a method comparison in healthy, young subjects. For the experimental setup in experiment 1, the measurement results were on average lower (200 Hz: 0.5 ± 0.4 µm, 30 Hz: 4.6 ± 2.3 µm) than for experiment 2 (200 Hz: 0.7 ± 0.4 µm, 30 Hz: 5.9 ± 3.1 µm), which is why we decided to consider the experiments separately with regard to stimulation effects.

## 3. Results

### 3.1. Subliminal Electrical Noise Stimulation and Vibration Perception Threshold in Experiment 1

In experiment 1, we found no statistically significant differences for VPT or CoV for either frequency between the conditions with and without SEN. Descriptively, mean VPT decreased at 30 Hz for the condition with SEN (VPT 30 Hz without SEN vs. 30 Hz with SEN: 93.5 ± 92.3 vs. 82.3 ± 78.5). VPT at 200 Hz did not change between the conditions (VPT 200 Hz without SEN vs. 200 Hz with SEN: 34.1 ± 22.1 vs. 34.7 ± 22.4) (Figure 2a,b). Furthermore, SEN led to decreased CoV for both measurement frequencies (CoV 30 Hz without SEN vs. 30 Hz with SEN: 0.25 ± 0.20 vs. 0.18 ± 0.13, CoV 200 Hz without SEN vs. 200 Hz with SEN: 0.21 ± 0.17 vs. 0.17 ± 0.18) (Figure 2c,d).

### 3.2. Subliminal Electrical Noise Stimulation and Vibration Perception Threshold in Experiment 2

The data from experiment 2 also showed no statistically significant differences for VPT or CoV for either frequency between the two conditions (Table 2).

## 4. Discussion

In this study, we investigated the effects of SEN stimulation on vibration perception in older adults with diabetes mellitus. We hypothesized an improved vibration perception following plantar SEN stimulation. Contrary to our hypothesis and to the literature on electrical noise stimulation in healthy younger and older adults [4,6,16,17], our measurement results did not demonstrate statistically significant effects of SEN stimulation on vibration perception.

### 4.1. Experiment 1 & 2

For experiment 1, no significant differences between the conditions with and without SEN stimulation were found. However, a descriptive analysis shows a trend-like mean reduction of VPT by about 12% under noise influence at the low measurement frequency (30 Hz). This trend may be due to the innervation of Meissner corpuscles by Aβ and C-fibers [31]. According to Faisal et al. [1], C-fibers are very sensitive to physiological channel noise. In contrast to thick, highly myelinated Aβ-fibers, they presumably respond more clearly to externally applied SEN stimulation. Since not all Meissner corpuscles exhibit this type of innervation, this may provide an explanatory model for the statistically non-significant study results at 30 Hz. Pacinian corpuscles, on the other hand, are innervated only by Aβ-fibers, which might explain why certain effects were not found at 200 Hz. It is possible that this fiber type is too well insulated by the myelin sheath and has too large of a diameter. However, it was recently shown that Pacinian corpuscles also have additional Aδ- and C-fibers [32]. Therefore, the reasons for the non-significant results at 200 Hz should be elicited in further studies. The survey of VPT is based on the subjective perception of the participants and is therefore associated with a high variability. Especially in patients with diabetes mellitus, this variability can be increased due to reduced sensation, formication, and pain in the legs/feet [33]. A previous study showed that similar stimulation with pink noise reduced the variance between the measurements trials [34]. Therefore, we also investigated potential effects of SEN stimulation on the detection accuracy of VPT using CoV. Although no inferentially statistical effects of stimulation were shown, the variability for detecting the vibration stimuli decreased descriptively by 4% at 200 Hz and by 7% at 30 Hz. This suggests, even if to a small degree, that the participants were able to sense the vibrations more consistently.

For experiment 2, VPT values at 200 Hz increased minimally following SEN stimulation. The VPT variability decreased under the influence of noise. For 30 Hz vibrations, the VPT remained the same for both conditions, and VPT variability increased slightly under SEN stimulation. However, since these differences did not reach significance, randomness of the measurement results cannot be excluded. Thus, contrary to our expectations, there were no statistically significant effects of SEN stimulation in experiment 2. The reasons for these findings are uncertain. Table 3 shows defining characteristics of both experiments and their advantages and disadvantages.

### 4.2. General Aspects

#### 4.2.1. Subliminal Electrical Noise Stimulation (SEN)

A possible explanation why no significant effects of SEN were detected lies in the SEN stimulation itself. As already described, the individual noise perception threshold was determined in reference to literature [4,6,19,20,21,22]. The used Digitimer DS5 is a certified medical device, which limits the applicable current range to a maximum of ±20 mA for safety reasons. In five participants from experiment 1 and one participant from experiment 2, the current was not perceived by the participants, even at ±20 mA. Therefore, we stimulated these participants with 90% of the maximum, which clearly underestimated their individual 90% noise perceptual threshold. However, this approach is not entirely consistent with the approach used to determine the actual noise perception threshold. It is known that SEN stimulation can enhance the perception of weak stimuli if the applied noise intensity has an ideal level [3,37]: a larger vibratory stimulus requires lower SEN stimulation and vice versa [3,38]. Therefore, the noise intensity may have been too low for our six participants. Furthermore, when using SEN, the noise perception threshold is highly variable and should be constantly monitored during prolonged use and readjusted if necessary [38]. Since the noise perception threshold was recorded immediately before the VPT measurement, this aspect can likely be excluded.

In our experiments, based on a study by Magalhães and Kohn [19], SEN stimulation with a bandwidth of 5–2000 Hz was used to stimulate both Meissner and Pacinian corpuscles and their innervating nerve fibers. While this research group found positive effects on balance control stimulating muscle receptors and Golgi tendon organs [15], it is possible that this range of the noise signal was inappropriate for our study. However, the applied bandwidths and noise signal forms are highly variable in the literature. For example, Plater et al. [16] used electro-tactile non-uniform white noise with a bandwidth of 0–50 Hz and different noise intensities to investigate the perception of 30 Hz vibrations on the hairy skin of the calf. Toledo et al. [17] employed a bandwidth of 5–1500 Hz and used Gaussian bandpass filtered noise with zero mean. According to Karpul and colleagues [38], the noise signal character affects the level of noise perception threshold. Thus, to investigate the effects of SEN stimulation on VPT, other characteristics of the noise signal, such as a different bandwidth, could be used in follow-up studies. In addition, the application to other areas of the body or directly to muscle [19], could be of interest.

#### 4.2.2. Biomechanical Skin Properties

To interpret our results, we considered additional characteristics of the participants to find potential explanatory models for our results. We considered the individual biomechanical skin characteristics of the participants as a starting point. In the course of this study, we collected skin hardness data using the Shore OO Durometer and skin thickness data using the handheld ultrasound device as described in [39]. In diabetes, increased blood glucose levels lead to thicker and harder skin [40]. The outer layer of the epidermis, the stratum corneum, forms more calluses than in healthy individuals [40] making it more difficult for the electrical current to flow into the foot due to the insulating properties especially of the stratum corneum [41]. Studies have shown that skin resistance can be reduced from ~2.5 kiloohms (kΩ) to ~500 Ω [42,43] or even to ~0 Ω by abrading the skin [42]. In our study, however, we did not assess to what extent and when the participants underwent podological treatment with corresponding callus removal prior to our measurements. Therefore, we suspected a relationship between current perception threshold and biomechanical skin properties. However, no correlation was found. Future studies should control for this aspect to exclude potential influencing factors.

#### 4.2.3. Neuropathy Deficit Score (NDS)

Another potential explanation for the absence of SEN effects lies in the relationship between the disease diabetes and VPT. Although it is generally well accepted that diabetes may lead to an impaired perception of plantar vibratory stimuli (e.g., [10,44]), a recent investigation showed that this is not necessarily the case [13]: it was shown that the severity of concomitant neuropathic deficits seems to play a more prominent role than diabetes alone [13]. Hence, we considered the Neuropathy Deficit Score (NDS) from experiment 1 and expected severe deficits to interfere with the effects of SEN stimulation, where disease-induced denervation [45] was already considered severe. When considering the upper tail of the NDS scores (severely affected patients with scores of 6–9, *n* = 6), VPT at 30 Hz clearly decreased following SEN stimulation. This marked reduction was not observed for the lower tail of the NDS scores (0–2, *n* = 6). Unfortunately, we only obtained NDS data from experiment 1. Our data suggest that only the most severely affected patients seemed to benefit from SEN stimulation (Appendix A) and further research is necessary to investigate this.

### 4.3. Limitations

The present study is limited in its significance regarding the number of participants and also due to the heterogeneous gender distribution. It has been proven that vibration sensitivity changes earlier in men with diabetes mellitus, and that male subjects show neuropathic changes more frequently than females [46]. Corresponding analyses of gender-related noise effects could not be performed because of the limiting factors mentioned above. Furthermore, we had no data on diabetes duration for some participants. However, based on the available studies, diabetes duration is associated to diabetes-related neurological decline [11,13,45]. Data from this study also suggest a correlation between diabetes duration and neuropathy severity.

We considered absolute VPT, which represent subjective perceptions at various parietal cortex areas. Importantly, vibratory afferent inputs already affect and modulate motor pathways at the spinal cord level [47,48], well before subjective perception. However, the extent to which the intervention may have contributed to the modulation of motor processes was not recorded. Future studies could examine the effect of SEN stimulation on motor activity. Similar to Najafi et al. [49], it would be conceivable to stimulate the entire foot to investigate relationships between stimulated sensory with motor processes of the human body, such as balance or gait.

### 4.4. Conclusions

Our results show only marginally present effects of SEN on thick, myelinated Aβ-fibers. On the basis of this study, it must be refrained from a clear clinical benefit of this stimulation form on vibration sensitivity. It is worth mentioning that the most severely affected individuals (neuropathy deficits) seem to benefit from SEN. Based on Faisal et al. [1], it is possible that SEN could be used for applications targeting pain and temperature perception (both mediated via C-fibers) [50]. However, these two sensory modalities were not investigated in the present study.

## Figures and Tables

**Figure 1 biomedicines-10-01880-f001:**
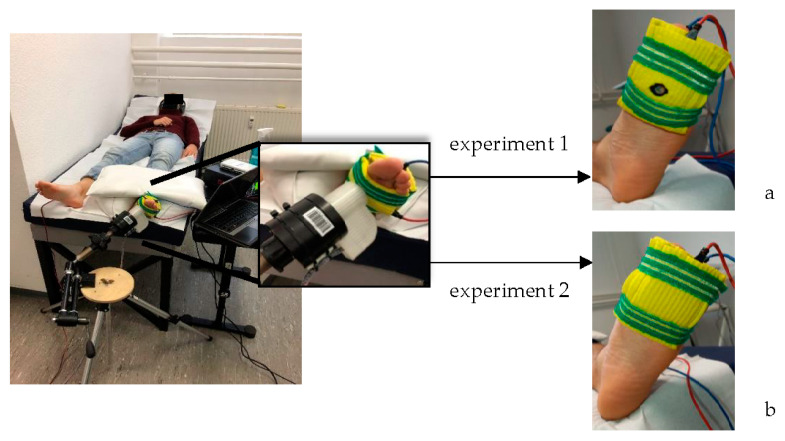
Setup of the VPT measurement with the applied plate electrodes for parallel subliminal electrical noise stimulation for experiment 1 (**a**) and experiment 2 (**b**).

**Figure 2 biomedicines-10-01880-f002:**
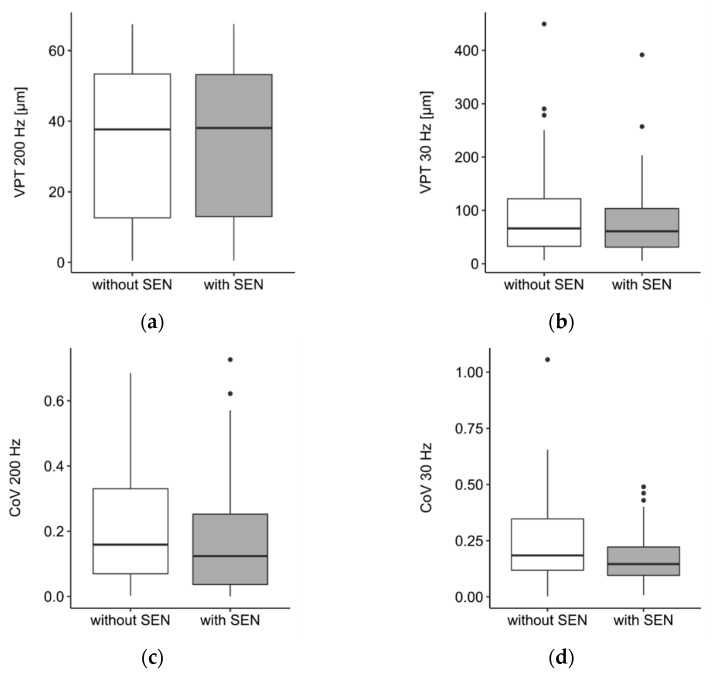
Boxplots showing vibration perception thresholds (VPT) at 200 Hz (**a**) and 30 Hz (**b**) and coefficient of variation (CoV) at 200 Hz (**c**) and 30 Hz (**d**) with and without subliminal electrical noise stimulation (SEN). Extreme values above 1.5 of the interquartile range are shown as dots.

**Table 1 biomedicines-10-01880-t001:** Anthropometric and clinical characteristics of the subjects, divided according to experiment.

	Age [years]	Height [m]	Mass [kg]	BMI [kg/m^2^]	Sex [m:f]	DD [years]	HbA1c [mmol/L]	Type [1:2]
experiment 1 (n = 38)	64.1 ± 9.6	1.7 ± 0.1	87.9 ± 17.7	30.1 ± 5.3	24:14	13.9 ± 10.5	7.2 ± 1.1 °	3:35
experiment 2 (n = 21)	67.9 ± 11.0	1.7 ± 0.1	90.0 ± 19.8	32.3 ± 6.2	5:16	10.1 ± 9.6 *	6.5 ± 2.0 *^,^°	0:19 *

Parameters are given as mean ± SD, except for sex (male to female ratio) and diabetes type (type 1 to type 2 ratio). DD: diabetes duration. * Missing values: diabetes duration was unknown in seven participants, HbA1c was unknown in two participants, diabetes type was unknown in two subjects, ° Statistically significant differences between experiment 1 and experiment 2.

**Table 2 biomedicines-10-01880-t002:** Mean ± SD VPT and CoV, differentiated according to the measurement condition at 200 Hz and 30 Hz.

	Parameter	200 Hz		30 Hz	
		without SEN	with SEN	without SEN	with SEN
Experiment 2	VPT [µm]	21.7 ± 21.3	22.4 ± 22.5	88.3 ± 67.9	88.4 ± 68.5
	CoV [µm]	0.21 ± 0.24	0.18 ± 0.16	0.14 ± 0.10	0.16 ± 0.12

SEN: subliminal electrical noise.

**Table 3 biomedicines-10-01880-t003:** Defining characteristics and their advantages (green) and disadvantages (red), differentiated by experiment. SEN: subliminal electrical noise stimulation.

	Experiment 1		Experiment 2
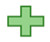	*direct detection of VPT* on the skin through a hole of 10 mm diameter in electrode and sponge	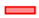	*indirect detection of VPT* through non-perforated electrode and sponge→ possible damping effect of sponge and electrode material on our results
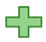	losing as little electrode and thus stimulation area as possible		
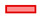	time-consuming positioning and readjustment of the probe to avoid spatial summation effects [35,36] at the 200 Hz measurements through contact between electrode and probe	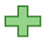	fast and uncomplicated positioning and readjustment of the probe towards the measurement location
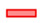	no guarantee that SEN actually flowed at the cut-out skin site	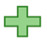	SEN flowed at the complete surface of the electrode and thus at the measurement location
	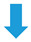		
**solution**	**adapted stimulation design for experiment 2**		

## Data Availability

The data presented in this study are available on request from the corresponding author. The data are not publicly available due to further analysis.

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
