# Peer review of "The Effect of Subliminal Electrical Noise Stimulation on Plantar Vibration Sensitivity in Persons with Diabetes Mellitus"

_biomedicines, 2022, doi:10.3390/biomedicines10081880_

Round 1

Reviewer 1 Report

Dear authors!

1. Please reduce the length of the Introduction which is also too broad and not straight to the point. I suggest a maximum of three paragraphs. 

2. In table 1, you should mention, whether there were any statistically significant differences between exp. 1 and exp. 2. 

3. "absence of diseases with a prognosis 90 of less than 5 years" please define, how this was determined. 

4. "peripheral arterial occlusive disease from stage two" please define the stage as this might not be clear to all the readers.

5. Discussion is too long. While reading, the reader loses focus. Please consider adding a table or find a clearer way to present the pros and cons and describe, why it came out negative. Describe possibilities for further research only briefly. And describe the clinical utility of findings or their lacking.  

Reviewer 2 Report

BIOMEDICINES-1817168 presents findings for NEL and VPT in persons with diabetes. Some comments for the authors to consider.

·         Title: Consider revising the title as “…in persons with diabetes mellitus” or similar.

·         Line 57: Use softer language (“causes”) here and throughout as needed.

·         Line 67: Avoid the use of “elderly” here and elsewhere. Instead consider “older adults”.

·         Lines 66-70 are not really needed. Consider revising the last paragraph to conclude with the purpose statement.  

·         Lines 88-101: How was this determined (self-report?)? Also, were there any exclusions after implementing the criteria?

·         The statistical analysis paragraph could be better identified with a sub-header and more detail in the text for how the analyses were executed. Some basic information is also missing such as statistical software used.

·         Line 264: Avoid re-introducing figures here and elsewhere in a Discussion for prospective flow.

·         Line 306: Avoid presenting a question in a Discussion.

·         Line 408: Avoid presenting results in a Discussion.

·         Make any changes to the abstract that align with those in the text.

Round 2

Reviewer 1 Report

The manuscript flows much better now. It is suitable for publication. Best regards!

Reviewer 2 Report

·         Section 2.1: Not sure if the placement for Table 1 is appropriate. At this point in the paper, the reader has no idea what type of statistical analyses were conducted within the table, nor should p-values be presented in the Methods. Revise and restructure for readability and prospective flow.

·         Discussion: A new section (Section 4.5) should be created so that the Conclusions are easily located after the limitations.
